# Enhanced immunocompatibility of ligand-targeted liposomes by attenuating natural IgM absorption

Juan Guan[1,2], Qing Shen[3], Zui Zhang[1,2], Zhuxuan Jiang[1], Yang Yang[1], Meiqing Lou[4], Jun Qian[2], Weiyue Lu[2] & Changyou Zhan[1,2]

Targeting ligands are anticipated to facilitate the precise delivery of therapeutic agents to diseased tissues; however, they may also severely affect the interaction of nanocarriers with plasma proteins. Here, we study the immunocompatibility of brain-targeted liposomes, which inversely correlates with absorbed natural IgM. Modification of long, stable positively charged peptide ligands on liposomes is inclined to absorb natural IgM, leading to rapid clearance and enhanced immunogenicity. Small peptidomimetic D8 developed by computer-aided peptide design exhibits improved immunocompatibility by attenuating natural IgM absorption. The present study highlights the effects of peptide ligands on the formed protein corona and in vivo fate of liposomes. Stable positively charged peptide ligands play double-edged roles in targeted delivery, preserving in vivo bioactivities for binding receptors and long-term unfavorable interactions with the innate immune system. The development of D8 provides insights into how to rationally design immunocompatible drug delivery systems by modulating the protein corona composition.

[1] School of Basic Medical Sciences & State Key Laboratory of Molecular Engineering of Polymers, Fudan University, Shanghai 200032, China. [2] School of Pharmacy, Fudan University & Key Laboratory of Smart Drug Delivery (Fudan University), Ministry of Education, Shanghai 201203, China. [3] State Key Laboratory of Oncogenes and Related Genes, Shanghai Cancer Institute, Renji Hospital, School of Medicine, Shanghai Jiao Tong University, Shanghai 200032, China. [4] Department of Neurosurgery, Shanghai General Hospital, School of Medicine, Shanghai Jiaotong University, Shanghai 200080, China. These authors contributed equally: Juan Guan, Qing Shen. Correspondence and requests for materials should be addressed to C.Z. (email: cyzhan@fudan.edu.cn)

Liposomes are clinically established as versatile drug delivery systems for several diseases, particularly cancer and infections[1]. Given their biocompatibility, biodegradability, and surface-tuning properties, stealth liposomes that achieve prolonged blood circulation by modifying hydrophilic polymers have attracted much interest in targeted drug delivery in the last few decades[2–5]. One strategy to achieve high targeting yield is to functionalize the surface of liposomes with targeting ligands (e.g., peptides, antibodies, or aptamers)[6,7]. These ligands are anticipated to facilitate precise delivery of therapeutic agents to diseased tissues by recognizing corresponding receptors or antigens.

Among different classes of targeting ligands, peptides are the objective of increasing scrutiny due, at least partially to their ease of synthesis and high throughput screening[8,9]. Peptide ligands exhibit high potency and specificity by occupying large interface of corresponding receptors[10,11]. Considerable efforts have been made to achieve enhanced targeting yields of peptide functionalized nanomedicines, such as by optimizing the structure of peptide ligands for high binding affinity and/or by stabilizing peptide ligands to overcome multiple enzymatic barriers in vivo[12–16]. However, the effects of peptide ligands on immunocompatibility of liposomes after modification remain elusive. In particular, after entry into blood stream, liposomes are immediately surrounded by high level of plasma proteins (or other biomolecules)[17]. They are associated within lipid surface to form a protein shell, referred to as "protein corona". The resulting biological identity may be far different from the pristine liposomes[18–21]. Modification of peptide ligands has serious impacts on the composition of the formed protein coronas, which determines the fate and transport of liposomes. For example, absorption of dysopsonins (e.g., albumin and apolipoproteins) prolongs blood circulation, while opsonins (e.g., immunoglobins and complements) induces rapid clearance of liposomes by the mononuclear phagocyte system (MPS)[22–26]. More attention should be paid to understand the rule and mechanism of interaction between plasma proteins and peptide functionalized liposomes for rational design of liposomes with high targeting yields and improved immunocompatibility in vivo.

In the present study, we re-interrogate the effects of peptide ligands on immunocompatibility of liposome-based, brain-targeted drug delivery systems. A pair of brain-targeted peptide ligands, LCDX and its retro-inverso peptide analog DCDX are selected and conjugated on the surface of stealth liposomes. The correlation between immunocompatibility and composition of formed protein coronas is deciphered. Modification of long, stable positively charged peptide ligands on liposomes induces enhanced absorption of natural IgM, which is attributed to the low immunocompatibility of DCDX-modified liposomes. The peptidomimetic D8 is rationally designed to achieve high brain-targeting capacity and good immunocompatibility by modulating the composition of protein corona.

## Results and discussion

**DCDX-modified stealth liposomes are immunogenic**. The use of peptidomimetics such as retro-inverso analogs has been described in targeted drug delivery[13,27,28]. In retro-inverso peptides, also called all-D retro or retro-enantio peptides, the side-chain orientation of the amino acid residues is retained while the direction of the peptide bonds is reversed by assembling D-amino acid residues in the reverse order with respect to the original sequence[29,30]. Retro-inverso peptide analogs are much more stable than natural peptides. They are fully resistant to proteolysis in blood circulation and tissue microenvironment to achieve enhanced targeting efficiency[31]. Given that stable peptides can serve as excellent synthetic antigens[32], the immunocompatibility

remains elusive of retro-inverso peptide analogs modified nanocarriers for targeted drug delivery. LCDX and its retro-inverso peptide analog DCDX possess brain-targeting property by nicotinic acetylcholine receptors (nAChRs)-mediated transcytosis[33,34]. DCDX consists of all-D amino acids thus being fully resistant to proteolysis; while LCDX is only stable in fresh mouse serum for minutes. Plain liposomes (sLip, containing 5% molar ratio of mPEG2000-DSPE without peptide modification), LCDX-modified liposomes (LCDX-sLip, containing 3% molar ratio of mPEG2000-DSPE and 2% molar ratio of LCDX-PEG3400-DSPE), and DCDX-modified liposomes (DCDX-sLip, containing 3% molar ratio of mPEG2000-DSPE and 2% molar ratio of DCDX-PEG3400-DSPE) were prepared using thin film hydration method (Methods section). To evaluate immunogenicity of sLip, LCDX-sLip, and DCDX-sLip, BALB/c mice received four doses (weekly) of liposomes containing the adjuvant lipid A through intraperitoneal injection. Blood was sampled 7 days after the fourth dose and antibodies were determined by ELISA using mPEG2000-DSPE (sLip), LCDX-PEG3400-DSPE (LCDX-sLip), or DCDX-PEG3400-DSPE (DCDX-sLip) as antigen. As shown in Fig. 1a, DCDX-sLip exhibited the highest immunogenicity among all liposomal formulations. After four doses, DCDX-sLip generated respective 100-fold and 50-fold higher IgGs than sLip and LCDX-sLip. In addition, DCDX-sLip (after four doses) induced much more anti-PEG IgM than LCDX-sLip and sLip (Fig. 1b, mPEG2000-DSPE as antigen for all formulations; Methods section).

Since dendritic cells induce primary immune responses in vitro and in vivo, they are important for studying immunogenicity[35]. Mouse bone marrow dendritic cells (BMDCs) were generated and characterized as previously described[36,37] (Fig. 2a). For antigen uptake, BMDCs were incubated with DiI-loaded liposomes (with fresh mouse serum, Methods section) for 4 h and the nuclei were stained with DAPI. As shown in Fig. 2b, c, DCDX-sLip (75.8%) exhibited the highest uptake by BMDCs, followed by LCDX-sLip (41.4%) and sLip (20.7%). DCs are highly effective at stimulating naive T cells in comparison to other antigen-presenting cells (APCs), such as B cells and macrophages. During DC activation, endosomal sorting and trafficking of lysosome is adjusted to favor peptide loading and surface display of loaded MHC class II-peptide complexes[38,39]. Even though antigen capture may be only half of the story, relatively high immunogenicity of DCDX-sLip may be partially attributed to enhanced uptake by DCs.

Recent studies have shown that lymph nodes (LNs) contain a large number of resident APCs, which are also capable of capturing and presenting antigen to T cells[40,41]. Mouse lymph nodes were isolated 12 h post intravenous injection and suspended in phosphate-buffered saline (PBS). APCs were labeled with anti-MHC II antibody and DiI+/MHC II+ cells were counted by flow cytometry (Fig. 2d). Although all liposomal formulations exhibited low-uptake efficiency (<10% APCs were DiI+) by LNs resident APCs, modification of both CDX peptides significantly increased uptake in comparison to sLip. DCDX-sLip and LCDX-sLip showed comparable uptake by in vivo LNs resident APCs. Given that L-peptide is extremely instable in the lysosomal compartments, it remains difficult to predict the presenting efficiency of LCDX by those LNs resident APCs.

RAW264.7 macrophages exhibited much higher phagocytic efficiency than DCs. More than 99% of cells were DiI+ after incubation with all DiI-loaded liposomal formulations. The intracellular fluorescence intensity of LCDX and DCDX-modified liposomes showed 2–2.5-fold increase in comparison to sLip (Fig. 2e, f). Pre-incubation with free DCDX peptide (200 μM) did not affect phagocytosis (Supplementary Fig. 1), indicating that CDX peptide modification may readily initiate non-specific phagocytosis and activate macrophages[42,43].

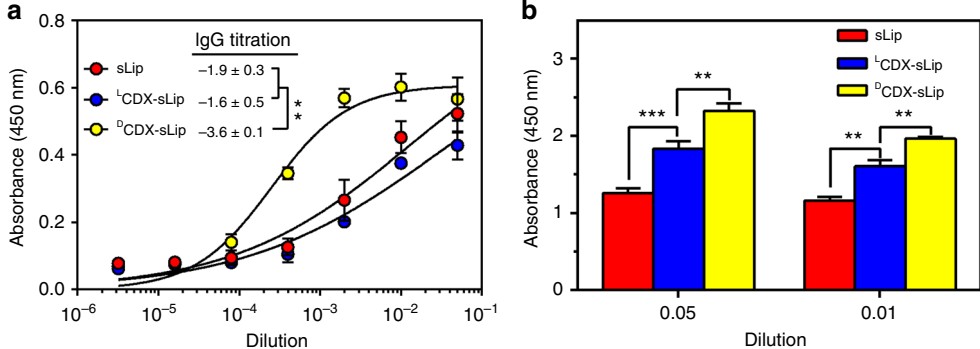

**Fig. 1** Immunogenicity of liposomal formulations. **a** IgG titrations. Absorbance in the ELISA plate versus serum dilution and antibody titer reported as log (EC50). mPEG2000-DSPE, $^L$CDX-PEG3400-DSPE, and $^D$CDX-PEG3400-DSPE were used as antigens for sLip, $^L$CDX-sLip, and $^D$CDX-sLip, respectively. **b** Absorbance in the ELISA plate for anti-PEG IgM evaluation. mPEG2000-DSPE was used as antigen for all formulations. $n = 3$, data are means ± s.d. Statistical significances were calculated by Student's $t$-test. **p < 0.01, ***p < 0.001

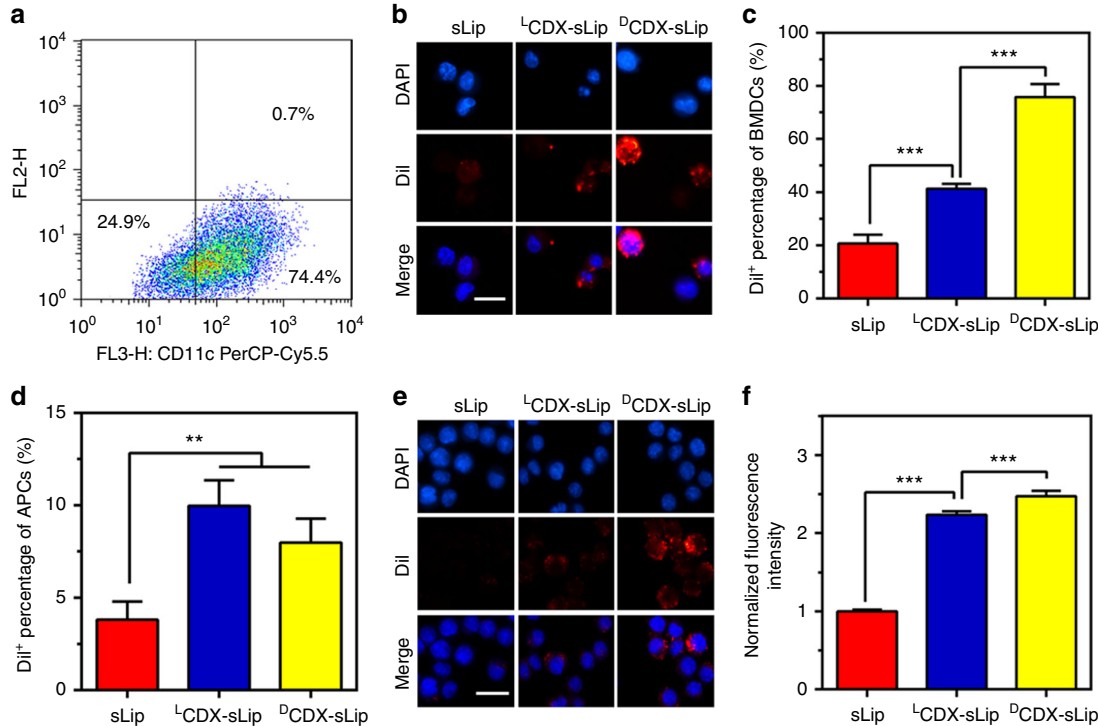

**Fig. 2** The uptake of sLip, $^L$CDX-sLip, and $^D$CDX-sLip by APCs. **a** The ratio of CD11c$^+$ cells in BMDCs. BMDCs were stained with anti-CD11c antibody and counted by flow cytometry. **b** Microscopy observation of DiI$^+$ BMDCs by confocal laser scanning microscopy. BMDCs were cultured with DiI-loaded liposomes for 4 h with serum and the nuclei were stained with DAPI. **c** The ratio of DiI$^+$ BMDCs counted by flow cytometry. **d** The ratio of DiI$^+$ cells in MHC II$^+$ APCs. Lymph nodes were isolated 12 h post injection, then suspended in PBS. Cell suspension was incubated with anti-MHC II antibody at 4 °C for 1 h and the ratio of DiI$^+$ cells in MHC II$^+$ APCs was counted by flow cytometry. **e** Microscopy observation of DiI$^+$ macrophages by confocal laser scanning microscopy. RAW264.7 cells were cultured with DiI-loaded liposomes for 1 h with serum and the nuclei were stained with DAPI. **f** The normalized fluorescence intensity in RAW264.7 cells by flow cytometry. Scale bar = 20 μm, $n = 3$, data are means ± s.d. Statistical significances were calculated by Student's $t$-test. **p < 0.01, ***p < 0.001

**$^D$CDX modification induces rapid liposome clearance.** Since CDX modification could induce enhanced uptake of liposomes by macrophages, it is very likely that CDX peptide modified liposomes are readily recognized by MPS. To study the pharmacokinetic profiles of liposomes, DiI-loaded liposomes were intravenously injected to BALB/c mice via the tail vein. At predetermined time points, mice were killed and the plasma concentration of DiI was measured (Methods section; Fig. 3a and

Supplementary Table 1). $^L$CDX-sLip and $^D$CDX-sLip exhibited rapid elimination in blood, decreasing the AUC$_{0-24}$ to 28% and 3% of sLip.

Accelerated blood clearance (ABC) effect of PEGylated liposomes has been reported by many groups, which is attributed to the generation of anti-PEG IgM (reaches the peak at 5–7 days) after a small dose of PEGylated liposomes[44,45]. To test the ABC effect, BALB/c mice were intravenously injected with a low dose

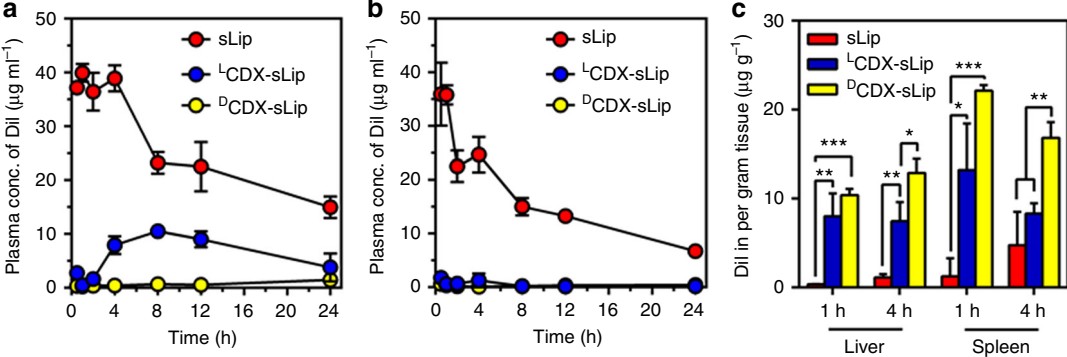

**Fig. 3** Pharmacokinetic profile and biodistribution of DiI-loaded sLip, $^L$CDX-sLip, and $^D$CDX-sLip. **a** Plasma concentration of DiI determined at 30 min, 1 h, 2 h, 4 h, 8 h, 12 h, and 24 h after intravenous injection in BALB/c mice. **b** The ABC effect of different liposomal formulations. BALB/c mice were injected with a low dose of empty liposomes (sLip, $^L$CDX-sLip, and $^D$CDX-sLip, 5 mg HSPC per kg of mouse), followed with a second injection of the normal dose of DiI-loaded liposomes 5 days after the first injection (50 mg HSPC per kg of mouse). **c** Biodistribution of liposomes in liver and spleen of BALB/c mice. $n = 3$, data are means ± s.d. Statistical significances were calculated by Student's t-test. *$p < 0.05$, **$p < 0.01$, ***$p < 0.001$

of empty liposomes (sLip, $^L$CDX-sLip, and $^D$CDX-sLip, 5 mg HSPC per kg of mouse), followed with a second injection of the normal dose (50 mg HSPC per kg of mouse) of DiI-loaded liposomes 5 days after the first injection. sLip exhibited moderate ABC effect in mice, demonstrating 40% decrease of $AUC_{0-24}$ (Fig. 3b and Supplementary Table 1). In sharp contrast, $^L$CDX-sLip and $^D$CDX-sLip only showed 2.5 and 2.1% $AUC_{0-24}$ of that of sLip after pre-dose triggering. CDX peptides modification exacerbated the ABC effect of liposomes, which may be explained by the enhanced anti-PEG IgM after repeated injections of both liposomal formulations (Fig. 1b).

To study the in vivo fate of liposomes, biodistribution of sLip, $^L$CDX-sLip, and $^D$CDX-sLip in liver and spleen of normal BALB/c mice was quantified (Fig. 3c). In the main metabolism (liver) and immune organ (spleen), both $^L$CDX-sLip and $^D$CDX-sLip exhibited much higher distribution than sLip 1 h and 4 h after injection. It is worthy of noting that at 4 h after injection, $^D$CDX-sLip exhibited much higher distribution in liver and spleen than $^L$CDX-sLip; while they displayed similar distribution level at 1 h after injection in both organs.

**Peptide stability affects protein corona composition.** Intravenously injected liposomes thoroughly interact with plasma proteins to form protein corona, and the composition of which closely relates to the surface properties of liposomes[46]. As for targeted drug delivery systems, the properties of targeting ligands, including stability, charge, size, and hydrophilicity, have major impacts on the composition of protein corona and in vivo fate of liposomes. All liposomes exhibited an average size of ~138 nm after extrusion through the 100 nm membrane (Supplementary Table 2). The modification of $^L$CDX and $^D$CDX increased the zeta-potentials of liposomes from −48 mV to −31 mV, which may be due to the net positive charges in both peptides (see peptide sequences in Supplementary Table 3).

To characterize the interaction between liposomes and plasma proteins, all liposomal formulations were incubated with 50% fresh mouse serum in vitro for 1 h at 37 °C, and the size and zeta-potential were measured without removal of the non-adherent plasma proteins (Methods section). Absorption of plasma proteins did not induce significant change of the particle size (Supplementary Table 2). However, the polydispersity index (PDI) increased significantly, which may be attributed to the absorption of plasma proteins and the mixing of large plasma proteins and plasma microvesicles. The zeta-potentials of all

liposomal formulations were increased from −20 to −15 mV after incubation with serum. The formed protein coronas were collected by centrifugation and rinsed with chilled PBS. The plasma protein pellets were separated using SDS-PAGE (Fig. 4a). Modification of CDX peptide ligands resulted in significant increase of a protein band at 72 kDa, which was ascertained as natural IgM by nano-LC-MS/MS.

Natural IgM exists in the circulation principally as a pentamer, and occasionally as a hexamer[47]. The unique structure of natural IgM allows it to interact with many other components of the immune system, including members of the complement system, mannose-binding lectin, and Fc receptor(s) for IgM[48,49]. Natural IgM has a key function in protecting against a range of viral, bacterial, fungal, and parasitic infections. With help from complement component C1q, natural IgM boosts their engulfment by phagocytes and increases the presentation of pathogen-derived antigens[50]. Thus, the increase of natural IgM absorption may be attributed to low immunocompatibility of $^D$CDX-sLip. $^L$CDX-sLip displayed comparable capacity of IgM binding in vitro, but were less immunogenic than $^D$CDX-sLip in vivo. To better understand the dynamical interaction of natural IgM with liposomes in vivo, protein coronas were collected at 1 and 4 h after intravenous injection of liposomes and the content of IgM was quantified using western blot assay (Fig. 4b, c, Methods section). Both $^D$CDX-sLip and $^L$CDX-sLip absorbed much more natural IgM at 1 h than sLip, indicating that CDX peptides on liposomal surface could rapidly interact with natural IgM in vivo. The content of IgM in the formed protein corona of $^D$CDX-sLip increased at 4 h compared with that at 1 h. On the contrary, the content of natural IgM in the formed protein corona of $^L$CDX-sLip at 1 h after injection was comparable to that of $^D$CDX-sLip; while it significantly decreased 4 h after injection. Given that $^L$CDX is subject to proteolysis in blood circulation[51], the present results suggested that instable peptide ligands may have highly variable composition of protein corona in vivo. It is interesting that plasma concentration of $^L$CDX-sLip rebounded at 4 h after injection (Fig. 3a), which may be attributed to the dynamic absorption of IgM. Stable ligands that can induce natural IgM absorption play double-edged roles in the in vivo fate of liposomes. The stability of ligands is crucial to targeted drug delivery systems, thus D-peptide ligands are effective to preserve the bioactivity of ligands in blood circulation. However, preservation of IgM absorption leads to rapid clearance and enhanced immunogenicity.

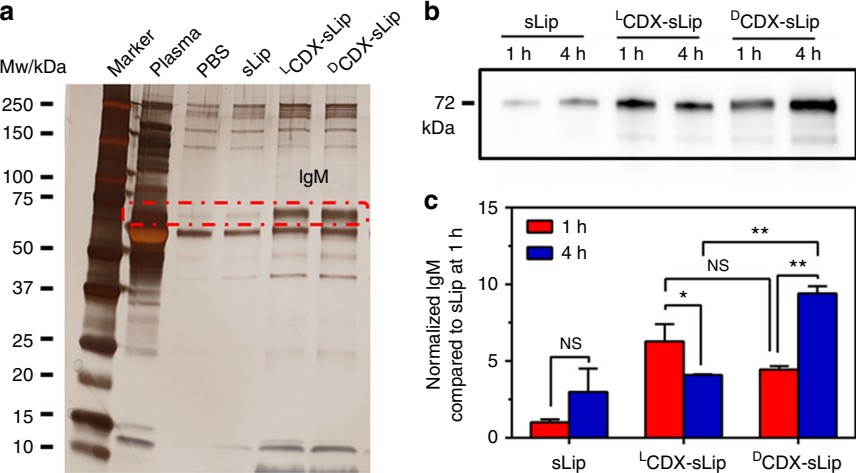

**Fig. 4** Effect of the modified peptides on the composition of protein corona. **a** Separation of protein corona by SDS-PAGE. IgM (at Mw 72 kDa, circled in the red dashed line) was characterized by nano-LC-MS/MS. **b** Western blot assay of natural IgM on liposomal surface in vivo 1 and 4 h after injection. **c** Quantification of absorbed natural IgM by normalizing the gray values. $n = 3$, data are means ± s.d. Statistical significances were calculated by Student's $t$-test. NS indicates not significant, $*p < 0.05$, $**p < 0.01$

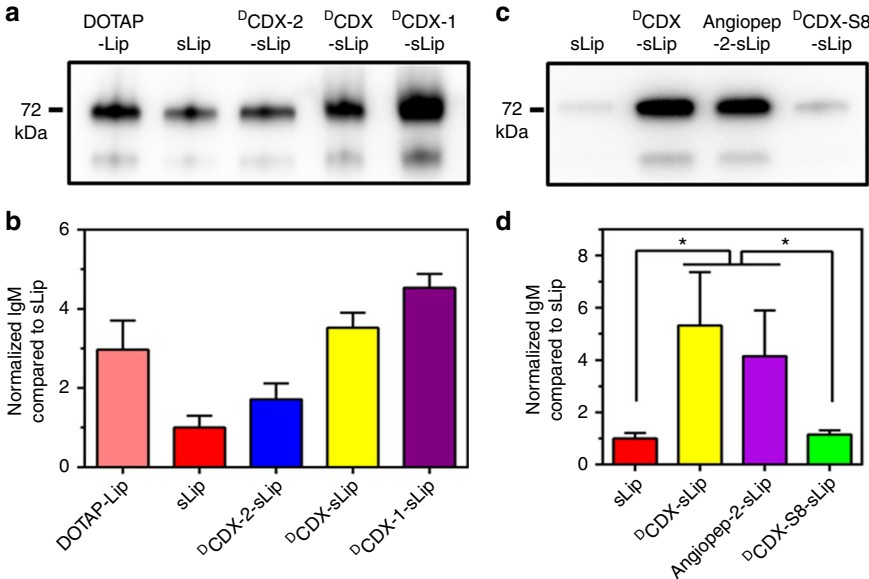

**Fig. 5** Effect of electrostatic interaction on the absorption of natural IgM. **a**, **c** Western blot assay of natural IgM on liposomal surface after incubation with serum for 1 h. **b**, **d** Quantification of absorbed natural IgM by normalizing gray values. $n = 3$, data are means ± s.d. Statistical significances were calculated by Student's $t$-test. $*p < 0.05$

**Electrostatic interaction dominates natural IgM absorption.** Both $^{L}$CDX and $^{D}$CDX are cationic, and the modification of CDX peptides increased zeta-potentials of liposomes to some extent (Supplementary Table 2). Since a crystal structure for IgM has not been resolved, prediction of the interaction modes between liposomes and natural IgM remains challenging. Electrostatic interaction has been reported to dominate the plasma proteins absorption on the surface of nanoparticles[52,53]. It is plausible that charge on liposomes may play important roles in absorption of natural IgM. Two $^{D}$CDX mutants, termed $^{D}$CDX-1 and $^{D}$CDX-2 (see sequences in Supplementary Table 3), were synthesized using solid-phase peptide synthesis. $^{D}$CDX has five positively charged residues ($^{D}$Arg2, $^{D}$Arg5, $^{D}$Arg8, $^{D}$Arg11, and $^{D}$Lys15) and three negatively charged residues ($^{D}$Glu3, $^{D}$Glu10, $^{D}$Glu14). In $^{D}$CDX-1, $^{D}$Glu3, and $^{D}$Glu14 were mutated with $^{D}$Ala, thus net positive charges increased to four. The net positive charges of $^{D}$CDX-2 decreased to zero after $^{D}$Arg2$^{D}$Ala and $^{D}$Lys15$^{D}$Ala mutations. $^{D}$CDX-1 and $^{D}$CDX-2 modified liposomes (containing 3% molar ratio of mPEG2000-DSPE and 2% molar ratio of peptide modified PEG-DSPE) were prepared (Supplementary Table 4) and their binding with natural IgM was quantified by western blot assay (Fig. 5a, c). As expected, the content of natural IgM in the formed protein corona positively correlated with the net positive charges of peptides ($^{D}$CDX-1-sLip > $^{D}$CDX-sLip > $^{D}$CDX-2-sLip). Natural IgM absorption on DOTAP-Lip (containing 5% molar ratio of DOTAP, (2,3-Dioleoyloxy-propyl)-trimethyl ammonium)

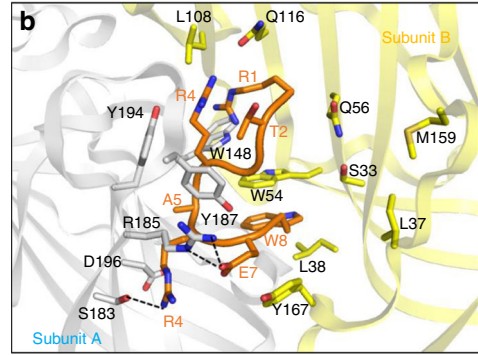

**Fig. 6** Binding mode of peptides with α7 nAChR. $^D$CDX-S8 (green, **a**) and D8 (brown, **b**) with α7 nAChR is shown. Subunit A of receptor is shown in white and subunit B is in yellow. Residues involved in binding are represented by sticks, and hydrogen bonds are denoted by black dashed lines

was also investigated. DOTAP-Lip exhibited much higher zeta-potential than $^D$CDX-1-sLip (Supplementary Table 4), while demonstrated less absorption of natural IgM on the surface. The positive charges of peptide are separated from the lipid bilayer by a PEG3400 spacer. In contrast, the positive charges of DOTAP are close to lipid bilayer. The present results suggested that the binding modes of plasma proteins on peptide modified stealth liposomes may differ from that on DOTAP-Lip.

Based on the binding mode of $^D$CDX with nAChR, N-terminal, and C-terminal four amino acids have no direct interactions with receptor[33]. To unveil the effect of peptide length on natural IgM binding, we synthesized a short D-peptide, termed $^D$CDX-S8, containing the middle eight residues that directly participate in receptor binding. $^D$CDX-S8 has three positively charged residues ($^D$Arg1, $^D$Arg4, and $^D$Arg7) and one negatively charged ($^D$Glu6). Interestingly, $^D$CDX-S8 modified liposomes ($^D$CDX-S8-sLip) exhibited a fourfold decrease of natural IgM absorption in comparison to $^D$CDX-sLip based on western blot assay (Fig. 5b, d). Angiopep-2 (19 residues, see sequence in Supplementary Table 3) is a widely used brain-targeting ligand, containing four positively charged residues (Arg8, Lys10, Arg11, and Lys15) and two negatively charged (Glu17 and Glu18). After 1 h incubation with serum, Angpep-2-sLip aggregated to some extent and exhibited huge PDI value (Supplementary Table 4). Angpep-2-sLip absorbed slightly less natural IgM than $^D$CDX-sLip, but much higher than $^D$CDX-S8-sLip. These results suggested that absorption of natural IgM positively correlates with not only the positive charge but also peptide length.

**Short peptidomimetic acquires enhanced immunocompatibility.** The aforementioned evidence suggested that enhanced absorption of natural IgM is, at least partially attributed to low immunocompatibility of $^D$CDX-sLip. Peptidomimetics that cause less natural IgM absorption may improve the immunocompatibility of targeted drug delivery systems. $^D$CDX-S8 provided a starting point for further optimization of short brain-targeting ligands with high affinity. Aided by computational peptide design, we analyzed the contribution of each residue in $^D$CDX-S8 to receptor binding using Rosetta peptide dock program[54]. In the binding mode of $^D$CDX-S8 (Fig. 6a, sequence number was kept as in $^D$CDX), $^D$Thr6, Gly7 and $^D$Ala9 did not form strong interactions with nAChRs. $^D$Arg8 extended into the cleft between subunit A and B, forming conserved cation–π interaction with Trp148 in subunit A. There were unfavorable interactions for $^D$Glu10 and $^D$Arg11 of $^D$CDX-S8. Ser183 and negative charged Asp196 in subunit A of nAChR were close to $^D$Glu10 of $^D$CDX-S8, and positively charged $^D$Arg8 was next to $^D$Arg11 of $^D$CDX-

S8. The side chain of $^D$Trp12 of $^D$CDX-S8 stretched deeply into the pocket, forming hydrophobic interaction with Trp54, Leu37, and Leu38 in subunit B of receptor. Based on the binding mode of $^D$CDX-S8, $^D$Glu10 was mutated to $^D$Arg in order to form hydrogen bond with Ser183 and/or electrostatic interaction with Asp196 in subunit A. $^D$Arg11 of $^D$CDX-S8 was mutated to Glu in order to form electrostatic interaction with Arg185 in subunit A. $^D$Thr2 was mutated to Thr for effective interaction with Tyr187. The binding mode of the peptidomimetic D8 ($^D$RTG$^D$R$^D$A-$^D$RE$^D$W) was examined (Fig. 6b). $^D$Arg1 formed cation–π interaction with Trp148 of subunit A. Thr2 interacted with Tyr187 by hydrophobic interaction. $^D$Arg4 formed cation–π interaction with Tyr194 and Tyr187. $^D$Arg6 interacted with Ser183 by hydrogen bonds and Glu7 interacted with Arg185 by electrostatic interaction. $^D$Trp8 stretched into the hydrophobic pocket formed by Trp54, Tyr167, and Leu38. The affinity of D8 was estimated by X-score program, exhibiting slightly stronger binding than $^D$CDX (Supplementary Table 5).

As expected, D8 demonstrated comparable stability in fresh mouse serum as $^D$CDX (Fig. 7a). To experimentally evaluate the affinity of D8, Neuro 2a cell line was used since it highly expresses nAChRs[55]. Serial dilutions of $^D$CDX and D8 were used to compete the binding of $^D$CDX-sLip/DiI with the nAChRs on Neuro 2a cells (Methods section), and DiI$^+$ cells were counted by flow cytometry (Fig. 7b). D8 and $^D$CDX could compete the binding of $^D$CDX-sLip/DiI with Neuro 2a cells in a dose-dependent manner, and both peptidomimetics displayed comparable efficiency. D8 was also conjugated on the surface of stealth liposomes (D8-sLip, Methods section), and the brain-targeting property was evaluated (Fig. 7c). In the in vitro brain capillary endothelial cell monolayer, D8-sLip showed comparable penetration efficiency to $^D$CDX-sLip, significantly higher than that of sLip. D8-sLip also displayed brain-targeting property in vivo (Fig. 7d). In the brain of BALB/c mice, both DiI-loaded $^D$CDX-sLip and D8-sLip demonstrated significant distribution at 4 h after intravenous injection. In contrast, liposomes without peptide modification (sLip) did not exhibit brain distribution. After 48 h incubation with bEnd.3 and AML12 cells, D8 peptide and D8-sLip did not demonstrate toxicity to both cell lines (Supplementary Figure 2).

The interaction of D8-sLip with plasma proteins was studied. The formed protein corona in vivo was separated by centrifugation and analyzed by western blotting, exhibiting that D8-sLip absorbed much less natural IgM than $^D$CDX-sLip (Fig. 8a, b). Subsequently, the pharmacokinetics and biodistribution of D8-sLip were studied in BALB/c mice. As shown in Fig. 8c and Supplementary Table 1, D8-sLip demonstrated 49% and 32% $AUC_{0-24}$ of that of sLip without or with a pre-dose of blank

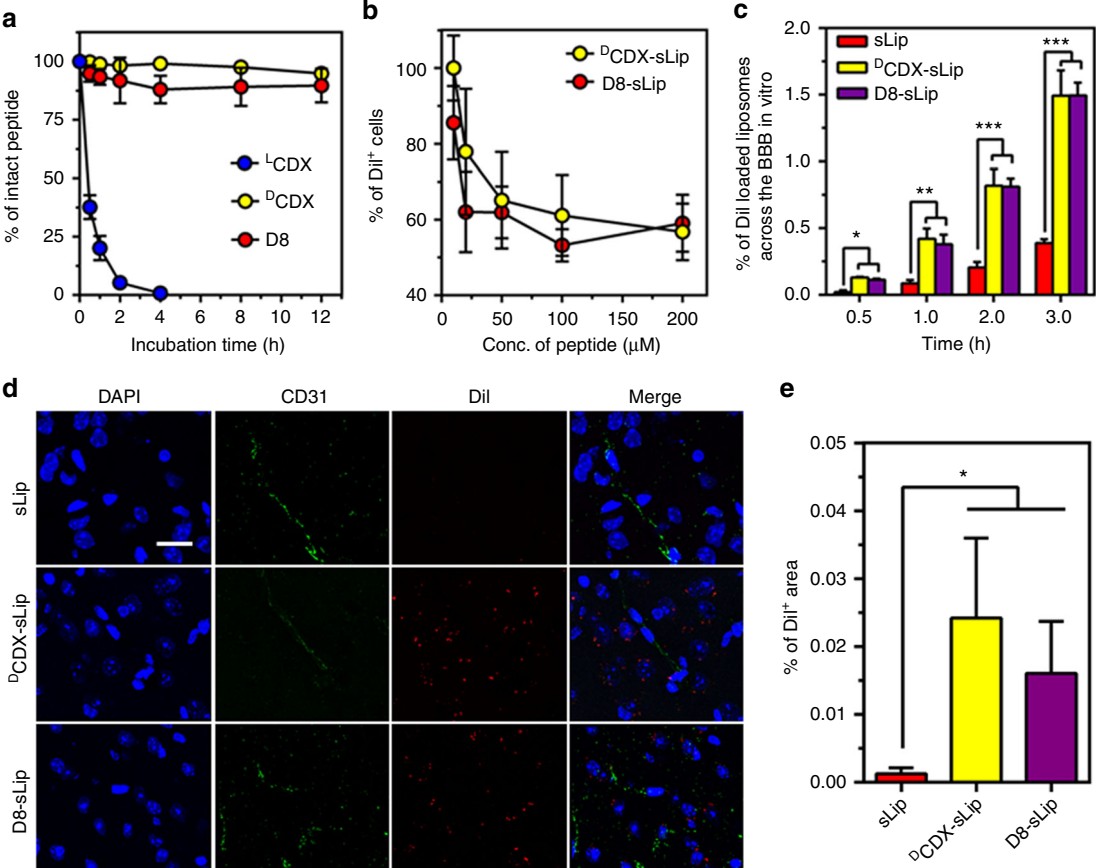

**Fig. 7** Characterization of D8. **a** Stability in mouse serum. D8, $^{L}$CDX, and $^{D}$CDX were dissolved in distilled water (1 mg mL$^{-1}$) and were incubated with mouse serum. RP-HPLC was used to monitor and quantify peptide hydrolysis at predetermined time points. **b** $^{D}$CDX and D8 competed the binding of DiI-loaded $^{D}$CDX-Lip with Neuro 2a cells. **c** Transcytosis efficiency of DiI-loaded sLip, $^{D}$CDX-sLip, and D8-sLip across the primary brain capillary endothelial cell monolayer. **d** Microscopic observation of brain distribution of liposomes. BALB/c mice was injected with DiI-loaded sLip, $^{D}$CDX-sLip, and D8-sLip via tail vein and brains were dissected, frozen sectioned, and stained with anti-CD31 antibody and DAPI 4 h post injection, blue—nuclei, green—blood vessels, and red—DiI dye, scale bar = 10 μm. **e** Positive DiI area in three random areas were counted by Image Pro. Scale bar = 20 μm, n = 3, data are means ± s.d. Statistical significances were calculated by Student's t-test.*p < 0.05, **p < 0.01, ***p < 0.001

liposomes. D8-sLip showed much less distribution in liver and spleen than $^{D}$CDX-sLip (Fig. 8d), suggesting that reducing natural IgM absorption could prolong blood circulation of liposomes by reducing the recognition of mononuclear macrophage system[56]. As expected, D8-sLip generated significantly less immunogenicity than $^{D}$CDX (Fig. 8e, f). These results suggested that enhanced immunocompatibility of targeted drug delivery systems could be achieved by attenuating natural IgM absorption, and the short peptidomimetic D8 presented a valuable ligand for efficient brain transport of drugs.

Altogether, net positive charges in peptide ligands with similar length showed good linear relationship with in vitro absorption of natural IgM (Fig. 9). Absorption of natural IgM negatively regulated the blood circulation of liposomes. These results verified that bulky absorption of natural IgM caused unfavorable immunocompatibility of liposomes. The successful development of D8-sLip revealed that enhanced immunocompatibility of ligand-targeted liposomes could be achieved by attenuating natural IgM absorption.

The effects of peptide ligands on liposome immunocompatibility have been investigated. Long, stable positively charged peptide ligands are prone to generate strong immunogenicity after modifying on liposomal surface, which is at least partially initialized by dendritic cells and lymph nodes resident APCs. They also cause enhanced phagocytosis by macrophages, leading

to rapid clearance in blood and heavy accumulation in liver and spleen. Analyses of formed protein corona on liposomal surface reveal that natural IgM absorption may play pivotal roles in stimulating in vivo unfavorable immunocompatibility. Short stable peptidomimetic ligand D8 has been developed by computer-aided peptide design, successfully preserving bioactivity in blood circulation and improving immunocompatibility of brain-targeted liposomes by attenuating natural IgM absorption. The present study brings insights into development promising ligand-targeted liposomes by precisely modulating the composition of formed protein corona in blood circulation.

## Methods

**Reagents and antibodies**. $^{L}$CDX, $^{D}$CDX, Angiopep-2, $^{D}$CDX-1, $^{D}$CDX-2, $^{D}$CDX-S8, and D8 were synthesized via solid-phase peptide synthesis using active ester chemistry to Fmoc-protected amino acid to the deprotected resin. Mal-PEG3400-DSPE, mPEG2000-DSPE, HSPC (hydrogenated soy phosphatidylcholine), and DOTAP (2,3-Dioleoyloxy-propyl)-trimethyl ammonium) were purchased from A. V.T. Pharmaceutical, Co., LTD. (Shanghai, China). DiI and lipid A were purchased from Sigma-Aldrich (St. Louis, MO). Cholesterol was purchased from Nippon Fine Chemical, Co., LTD (Takasago, Japan). Fast silver stain kit was from Beyotime Biotechnology (Jiangsu, China). Horseradish peroxidase labeled anti-mouse IgM antibody (Cat# ab97230, 1:5000), FITC labeled anti-MHC II antibody (14-4-4 s, Cat# ab25023, 1:200), Cy5.5 labeled anti-CD11c antibody (N418, Cat# ab210308), and anti-CD31 antibody (ab28364, 1:50) were purchased from Abcam (Cambridge, MA).

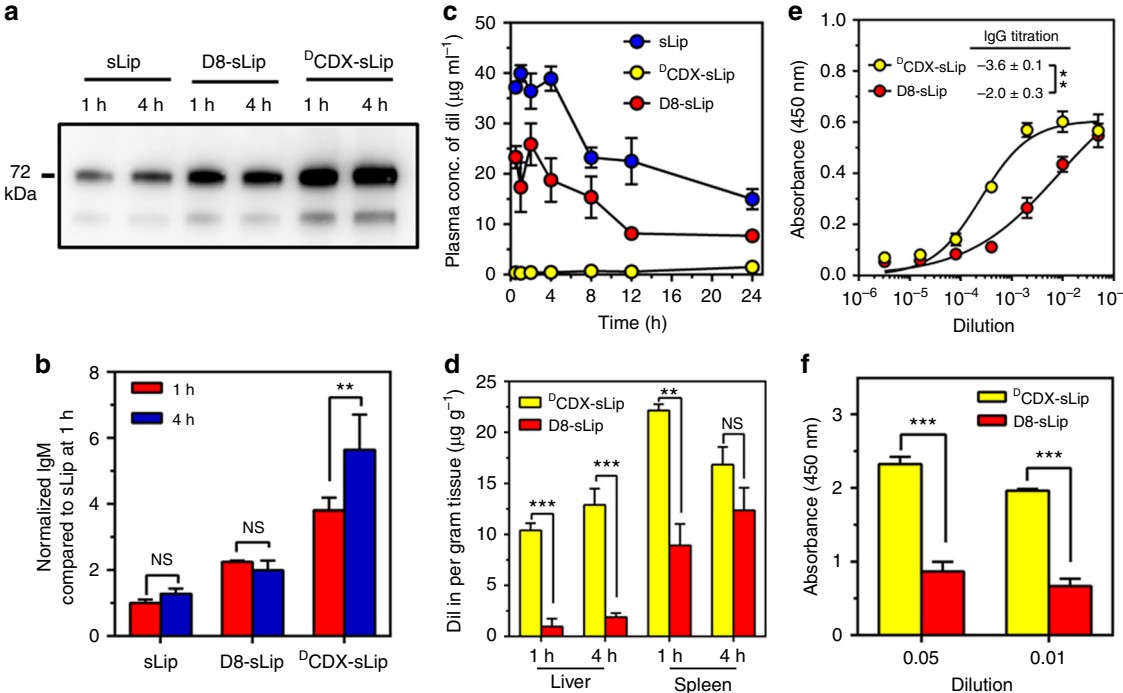

**Fig. 8** Immunocompatibility of D8-sLip. **a**, **b** Western blot assay (**a**) and quantification (**b**) of natural IgM on liposomal surface in vivo 1 and 4 h after injection. **c** Pharmacokinetic profiles of sLip, D8-sLip, and $^{D}$CDX-sLip. **d** Biodistribution in liver and spleen of BALB/c mice of D8-sLip and $^{D}$CDX-sLip. **e** IgG titrations. Absorbance in the ELISA plate versus serum dilution and antibody titer reported as log (EC50). D8-PEG3400-DSPE and $^{D}$CDX-PEG3400-DSPE was used as antigens for D8-sLip and $^{D}$CDX-sLip, respectively. **f** Absorbance in the ELISA plate for anti-PEG IgM evaluation. mPEG2000-DSPE was used as antigen for both formulations. $n = 3$, data are means ± s.d. Statistical significances were calculated by Student's $t$-test. NS indicates not significant, *$p < 0.05$, **$p < 0.01$, ***$p < 0.001$

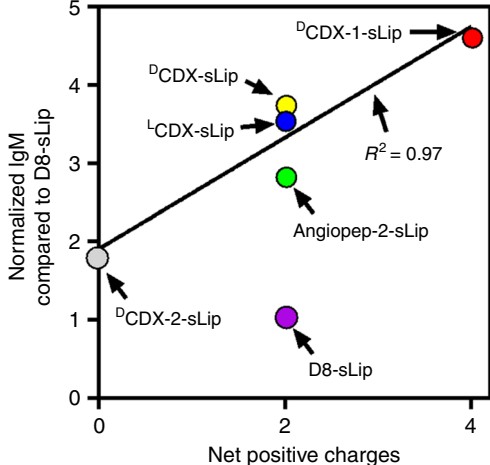

**Fig. 9** Relationship between net positive charge of peptide ligands and absorbed natural IgM. $n = 3$, data are means. Linear regression was performed with $^{L}$CDX-sLip, $^{D}$CDX-sLip, $^{D}$CDX-1-sLip, and $^{D}$CDX-2-sLip

**Cell lines and mouse models**. RAW264.7, Neuro 2a, AML12, and bEnd.3 cell lines were purchased from the Type Culture Collection of Chinese Academy of Sciences (Shanghai, China). Adult male BALB/c mice with an age of 6–8 weeks were purchased from Shanghai SLAC Laboratory Animal Co., Ltd (Shanghai, China) and kept under SPF condition. All animal experiments were carried out without blinding and in accordance with guidelines evaluated and approved by the ethics committee of Fudan University.

**Preparation and characterization of liposomes**. All peptides (containing an additional cysteine in the N-termini) were conjugated with Mal-PEG3400-DSPE

via sulfhydryl-maleimide coupling method[57]. In brief, Mal-PEG3400-DSPE was diluted in chloroform and rotary evaporated to form a thin film, then dried for 2 h under vacuum and hydrated with pure water at 37 °C. Peptides were dissolved in 0.1 M phosphate-buffered solution and mixed with Mal-PEG3400-DSPE (peptides/Mal-PEG3400-DSPE, molar ratio 1.5:1) to react for 6 h at room temperature. The mixture was placed in dialysis bag with molecular weight cutoff of 8000–1400 Da for 48 h to remove the residual free peptides and freeze-dried. The 1H-NMR spectra were collected (Supplementary Fig. 3). The disappear of the characteristic peak of maleimide group at 6.8 ppm in the 1H-NMR spectra of all peptide-PEG3400-DSPE indicated the successful reaction[58,59]. Liposomes were prepared by the thin film hydration and extrusion method[60]. For blank liposomes, a mixture of HSPC/cholesterol/mPEG2000-DSPE/peptides-PEG3400-DSPE (molar ratio of 52:43:3:2) or HSPC/cholesterol/DOTAP (DOTAP-Lip, molar ratio of 52:43:5) in CHCl₃ solution was rotary evaporated to form a thin film, then dried overnight under vacuum. The dried lipid film was hydrated with phosphate buffer saline at 60 °C. The lipid dispersion was extruded through polycarbonate membranes with a pore diameter of 400 nm, 200 nm, and 100 nm. DiI- or lipid A-loaded liposomes were prepared following the same methods as aforementioned, expect that DiI or lipid A was added before formation of thin film (0.4 mg DiI per 7.85 mg HSPC; 20 µg lipid A per 7.85 mg HSPC). The size and zeta-potential of liposome was detected in deionized water at a lipid concentration of 0.125 mM using ZetasizerNano ZS90 (Malven Instrument, southborough, MA). To detect the effects of serum on size and zeta-potential, liposomes in deionized water were incubated with 50% fresh mouse serum at a final lipid concentration of 0.125 mM for 1 h at 37 °C before measurement.

**Evaluation of stability of DiI-loaded liposomes**. The stability of DiI-loaded liposomes was evaluated by detecting fluorescence intensity of DiI after 4 h incubation with mouse serum at 37 °C. The liposomes were diluted by PBS containing 1 mM octyl-β-D-glucopyranoside centrifuged at 14,000×g for 30 min. The concentration of DiI in supernatant was measured by a fluorescence spectrophotometer. Liposomes destroyed by octyl-β-D-glucopyranoside (100 mM, 2:1 volume ratio to liposomes) were set as 100% destruction[61]. After sonication the mixture was diluted with PBS by 100 folds and centrifuged at 14,000×g for 30 min. The baseline of fluorescence intensity of liposomes was set as direct dilution by PBS containing 1 mM octyl-β-D-glucopyranoside (0% destruction of liposomes). The results indicated that less than 0.5% of liposomes were instable after 4 h incubation with mouse serum.

**Immunogenicity of the liposomes**. To evaluate immunogenicity of sLip, [L]CDX-sLip, [D]CDX-sLip, and D8-sLip, BALB/c mice received four doses (weekly, 50 mg HSPC per kg of mouse) of liposomes containing lipid A through intraperitoneal injection. Blood was sampled 7 days after the fourth injection and plasma was collected after centrifugation at $1000 \times g$ for 10 min.

IgG antibodies were determined by ELISA using mPEG-DSPE, [L]CDX-PEG-DSPE, or [D]CDX-PEG-DSPE as antigen. Microtiter wells were coated with antigen (2 μg per well) overnight. Wells were rinsed with PBS and blocked with 1% BSA for 1 h. Blood samples were serially diluted with PBS and incubated in the microtiter wells for 1 h. After thrice rinses with PBS, goat anti-mouse IgG antibody conjugated to horseradish peroxidase was added to react with IgG. 3,3′,5,5′-Tetramethylbenzidine (TMB) was added for 5–15 min and 2 M sulfuric acid was used to terminate the reaction. UV absorption was detected at 450 nm. Anti-PEG IgM was determined using the similar procedure as IgG, except that only mPEG-DSPE was used as antigen for all formulations.

**BMDCs culture and antigen uptake**. Bone marrow cells were isolated from BALB/c mice, and cultured in 1640 medium with FBS (10%), GM-CSF (20 ng mL$^{-1}$), and IL-4 (20 ng mL$^{-1}$) at 37 °C for 7 days. When the percentage of CD11c$^+$ cells are over 70% by flow cytomertry, BMDCs could be used in the following experiments. For antigen uptake, BMDCs were cultured with DiI-loaded liposomes with serum at 37 °C for 4 h. Cells were harvested and the nuclei were stained with 4′,6-diamidino-2-phenylindole (DAPI). Images were captured by a fluorescence microscope. The ratio of DiI$^+$ cells was counted by flow cytometry.

**RAW264.7 cells stimulation and antigen uptake**. RAW264.7 cells were cultured in DMEM medium containing 10% FBS with 5% CO$_2$ at 37 °C. Cells in exponential growth period were seeded in cell culture plate. After 12 h incubation, FBS was removed and DiI-loaded liposomes pre-incubated with mice serum for 1 h at 37 °C were added. After additional 1 h incubation, cells were harvested and the nuclei were stained with DAPI. Images were captured by a fluorescence microscope. The ratio of DiI$^+$ cells was counted by flow cytometry. To exclude specific interaction between CDX-modified liposomes and macrophages, RAW264.7 cells were pre-incubated with free [D]CDX peptide (200 μM) for 2 h before the liposomes were added and other operation were consist with the aforementioned.

**Liposome uptake by mouse APCs in lymph nodes**. BALB/c mice were intravenously injected with DiI-loaded liposomes (50 mg HSPC per kg of mouse) and killed 12 h after injection. Axillary and popliteal lymph nodes were harvested. Single cell suspensions were prepared and incubated with FITC-anti-mouse MHCII antibody at room temperature for 1 h. The uptake of different liposomes by MHC II$^+$ APCs was quantified by flow cytometry.

**Pharmacokinetic study**. To investigate the pharmacokinetic profile of liposomes, BALB/c mice were intravenously injected with DiI-loaded liposomes (50 mg HSPC per kg of mouse) and killed at 30 min, 1 h, 2 h, 4 h, 8 h, 12 h, and 24 h. Blood was collected and plasma was separated by centrifugation at $1000 \times g$ for 8 min. The plasma concentration of DiI was measured by a fluorescence spectrophotometer (Ex at 550 nm and Em at 570 nm).

To study the ABC effect, BALB/c mice were injected with a low dose of empty liposomes (sLip, [L]CDX-sLip, and [D]CDX-sLip, 5 mg HSPC per kg of mouse), followed with a second injection of the normal dose of DiI-loaded liposomes 5 days after the first injection (50 mg HSPC per kg of mouse). The blood was sampled and analyzed as aforementioned.

**Biodistribution study**. To quantify the biodistribution, BALB/c mice were intravenously injected with DiI-loaded liposomes (50 mg HSPC per kg of mouse). Blood was sampled and mice were killed at 1, 4, 12, and 24 h after injection. Liver and spleen were collected, weighed and DiI was extracted using methanol[25]. After the tissues were weighed and transferred into a 2 mL tube, 0.5 mL triton-X 100 (with a percentage of 5% in water) was added and the tissues were smashed by an ultrasonic pulverizer. With 500 μL methanol to extraction the DiI in tissues and separated by high speed centrifugation. Fluorescence intensity of plasma and tissue extraction was detected by a fluorescence spectrophotometer (Ex at 550 nm and Em. at 570 nm).

**Characterization of protein corona**. Whole blood was collected from male BALB/c mice, kept at room temperature for 30 min and centrifuged at $1000 \times g$ to separate serum. One hundred microliters plain liposome (with equal amount of phospholipid) in PBS was incubated with the same volume serum at 37 °C for 1 h. The mixture was centrifuged at $14,000 \times g$ for 30 min, and the pellet was rinsed with cold PBS (300 μL) twice. A plasma aliquot was subject to the same procedure as control. The pellet was boiled in a solution containing 5 μL SDS-PAGE 5× sample buffer, 20 μL PBS and 2 μL β-mercaptethanol at 100 °C for 10 min. Two microliters mouse serum was loaded as control. Electrophoresis was performed using gradient polyacrylamide gel, which was stained with Fast Silver Stain Kit. The bands were

cut and destined, reduced and alkylated. After trypsin digestion, the digested peptides were analyzed by nano-LC−MS/MS on an LTQ Orbitrap Fusion mass spectrometer (Thermo Electron, San Jose, CA).

To characterize in vivo natural IgM absorption on liposomal surface, BALB/c mice were intravenously injected with DiI loaded liposomes (50 mg HSPC per kg of mouse) and anesthetized at 1 h and 4 h. Blood was sampled and the serum was collected after centrifugation. The fluorescence intensity of the blood serum was measured by a fluorescence spectrophotometer to quantify the concentration of DiI-loaded liposome (Ex at 550 nm and Em at 570 nm). Serum was centrifuged at $14,000 \times g$ for 30 min to pellet the liposome-protein complex, which was denatured as aforementioned. Electrophoresis was carried out using gradient polyacrylamide gel as the separating gel. After western transfer, the PVDF membrane was incubated with anti-mouse IgM antibody for 10 h at 4 °C.

**Characterization of D8**. D8 was dissolved in distilled water (1 mg mL$^{-1}$). 0.1 mL of each was incubated with 0.9 mL 25% sterile rat serum. After 0.25, 0.5, 1, 2, 4, 8, and 12 h incubation at 37 °C, 20 μL acetonitrile (0.1% TFA) was added into 100 μL reaction mixture. The mixture stored at 4 °C for 20 min and centrifuged at $14,000 \times g$ for 10 min. An aliquot of 20 μL supernatant was analyzed by RP-HPLC to monitor and quantify peptide hydrolysis. To experimentally evaluate the affinity of D8, [D]CDX, and D8 peptides were serially diluted in PBS (10 μM, 25 μM, 50 μM, 100 μM, and 200 μM) and pre-incubated with Neuro 2a cells for 2 h at 4 °C, then [D]CDX-sLip/DiI was added to competitive binding with nAChRs on Neuro 2a overnight. Cells were rinsed with PBS and DiI$^+$ cells were counted by flow cytometry.

**Safety evaluation of D8 peptide and D8-sLip**. AML12 and bEnd.3 cell lines were seeded into 96-well plate. D8 peptides, sLip, and D8-sLip were serially diluted in DMEM and incubated with cells for 48 h. Cell viability was measured using MTS reagent.

**Brain transport efficiency**. Rat primary brain capillary endothelial cells (BCECs) were isolated and seeded on transwell chamber according to previous report[62]. The formed in vitro BBB monolayer was used for the following experiments once the transendothelial electrical resistance (TEER) was >250 Ω cm$^{-2}$. DiI-loaded different liposomes (5 μg mL$^{-1}$ of DiI) in DMEM containing 10% FBS was placed in the upper chamber. At 0.5, 1, 2, and 3 h, 200 μL medium from lower chamber was sampled and supplemented with the same volume of fresh medium. The fluorescence intensity of the sampled medium was detected by a fluorescence spectrophotometer (Ex. at 550 nm and Em. at 570 nm).

**Brain distribution in vivo**. BALB/c mice were intravenously injected with DiI-loaded sLip, [D]CDX-sLip, or D8-sLip. The brains were dissected 4 h after injection, and frozen sectioned and stained with anti-CD31 antibody and DAPI for microscopic observation and quantification by Image Pro software of the distribution of liposomes.

**Statistical analysis**. Data are presented as the means ± standard deviations (SD) and analyzed by Student's $t$-test with GraphPad Prism software 6.5.0. $p < 0.05$ was considered statistically significant (NS: $p > 0.05$, $0.01 < $ *$p < 0.05$, $0.001 < $ **$p < 0.01$, ***$p < 0.001$).

**Date availability**. The data that support the findings of this study are available within the paper and the Supplementary Information. All other data are available from the authors upon reasonable request.

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

## Acknowledgements

This work was supported by National Natural Science Foundation of China (81673361, 81690263, and 81673370), Shanghai Education Commission Major Project (2017-01-07-00-07-E00052), Shanghai Natural Science Foundation (18ZR1404800), and the State Key Laboratory of Molecular Engineering of Polymers (Fudan University).

## Author contributions

C.Z. and W.L. conceived and designed the research. J.G., Z.Z., Z.J., Y.Y., M.L., and J.Q. performed experiments. Q.S. designed the short peptidomimetic D8. C.Z. and J.G.

analyzed the data and wrote the manuscript. All authors read and approved the manuscript.

## Additional information

**Competing interests:** The authors declare no competing interests.

