## [Peer Review File · Nature Communications]

Reviewers' comments:

Reviewer #1 (Remarks to the Author):

This article describes a methodological approach to address immunocompatibility issues of surface-modified liposomes and design liposomes which are more immunocompatible. By studying the interaction of CDX peptide-modified liposomes in a step-by-step manner, the authors have been able to convince this reviewer with the data and interpretation. I have no hesitation in congratulating the investigators for this excellent study that too after a long time in the liposome field. Having said this, there are a few concerns that need attention to improve this work.

The only major regret is the absence of in vivo ABC data which would provide confirmation about the better immunocompatibility of the final D8-liposomes.

1. Include a mention of route of administration in Results and Discussion section. This information will help reader.
2. Only phagocytic activity of dendritic cells and macrophages has been shown. Whether these cells are immune-activated after incubation with liposome preparations is not shown.
3. The authors should discuss the possibility that the CDX-modified liposomes may be interacting with cells with specific interaction. The use of competing free peptide or somehow justifying the absence of expression of a receptor for CDX on the cells used is required. It is notable that macrophages have been shown to express nicotinic receptors.
4. Caveat for section of Electrostatic interaction...": The authors show that number of +ve charges are a factor, and so is the length of the peptide. But it is also possible that the position of a charge is also material, especially in specific interactions. In a way this caveat is proven to exist by the success of modified short D8 peptide where the authors manipulated the placement of charged amino acids.
5. PEG is no longer the only liposome-modifying polymer for stealth property. A mention of newer polymers should be mentioned in the introduction section. For instance, superhydrophilic polymers (Nag et al J Pharm Sci 2015). A statement on how these new developments would impact the conclusions drawn on the basis of this study is important.

Minor comments:

1. In methods section, authors mentioned lipid A in the composition of liposomes. Why?
2. Figure 2c: Y axis title spelling of across.

Reviewer #2 (Remarks to the Author):

This is a very interesting paper on a very well-known problem that limits the potential of targeted drug delivery. The protein corona has been indeed studied by many groups and for many years, however the solution to the problem and the way to effectively manipulate the type of proteins that will be adsorbed on the surface of nanoparticles immediately after their appearance in the blood in order to preserve the targeting capability of the ligands and the stealth nature of nanoparticles (and maximize immune-compatibility) has not been yet accomplished.

In this paper the main goal is brain targeting using ligand-targeted liposomes. The authors start with methodologies to correlate the properties of ligands with the types of proteins that are adsorbed on liposomes and their effects on the liposome immunocompatibility and blood circulation. After that they continue by using the previous conclusions, to design specific peptide ligands for brain targeting and carry out final tests to identify the targeting to the brain. However, there are several points that are missing or not discussed and in several instances the experimental setup, may not be optimal.

To be more specific:

1. The authors herein carry out a series of experiments to initially prove that the in vivo stability of a peptide ligand is important in order to be able to correlate the liposome pharmacokinetics with the IgG produced following liposome in vivo administration, and IgM adsorption of the surface of the liposomes (Figures 1-4). However they do not really explain what is happening in the case of the unstable peptide-bearing liposomes. This reviewer believes that a more in depth analysis of the particular results of the LCDX-liposomes may provide interesting insights about other aspects that may be important; but have not been considered, or have been overlooked.

2. For the previous they use control liposomes sLIP, consisted of HSPC/Chol and DSPE-PEG2000, and liposomes having on their surface L-CDX (unstable) peptide or D-CDX (stable peptide). There are a few important points about the liposomes, I would like to point-out: A) First of all the zeta potential of HSPC/CHOL/PEG liposomes is highly negative (Table 1), although there is no negatively charged lipid in their membrane (HSPC is a zwitterionic lipid). The only possible explanation for this may be the partial hydrolysis of HSPC to Lyso-PC (which is known to happen after 1 or 2 weeks, however PEG is known to shield the negative charge generated by hydrolysis of HSPC – see *Biochimica et Biophysica Acta* 1818 (2012) 2801–2807) B) The “stealth” character of the sLip is not obvious from the pharmacokinetic profile presented in Fig. 3.a. At the 4 h time point, although there is about half of the initial Dil level remaining in the blood, there is almost no Dil in the liver. Where is the Dil?? C) The peptides are attached on the liposomes via maleimide-linked on DSPE-PEG3500, while the control liposomes contain only PEG-2000, Can this difference be connected with the anti-PEG-IgM levels of the LCDX-sLIP compared to the control sLIP [although the IgG levels are close for these two liposomes types (which is logical since the L-peptide should be hydrolysed in a few minutes in the blood).?] Additionally, it is not mentioned if the attachment yield of the different ligands is measured and how. Are all the ligands attached by 100% (perhaps this is the case if ligand-PEG-lipids are synthesized initially – however it would be good to also measure the amount of ligand on the various liposome types) D) Through the full manuscript, all the results associated with the physicochemical properties of the liposomes (Table 2 and 3) are reported without SD values, although it is mentioned that n=3. This makes it difficult to understand the significance of specific differences between different types of liposomes. Furthermore, the authors do not comment at all about the very dramatic increase of the PDI values when the liposomes are measured after serum incubation. If the size presented is the mean size, I wonder if there is just one peak or several (due perhaps to aggregated liposomes) in the analytical report of the Malvern sizer. Also how were non-adherent proteins removed from the liposomes? Since the liposomes are small, it would be difficult to separate them from large serum proteins by ultracentrifugation. E) the high level of IgM after 1 h on LCDX-sLip is very strange and no explanation is given; results are more logical after 4 h. Do the authors have any information about the integrity of the different liposomes they are studying after 1 or 4 h in presence of serum proteins? Are they sure they are still intact and nano-sized? Are they aggregated? Is the Dil still associated with the liposomes?

3. After the first part of the study, the authors investigate the effect of the net charge of the peptide ligands on the electrostatic interaction with natural IgM, and show that a specific ligand CDX-S8 that has a specific amount of net charge, results in minimal interaction with IgM, compared to other liposome types; based on this, they then prepare a new small peptide as a ligand for brain targeting. However, in this experiment (presented in Figure 5) some of the liposomes used are very different in order to make any solid conclusions (case of DOTAP), so I don't understand why they were selected initially. Also, it is confusing that the results are presented differently in Figures 5. b and d. How does CDX-S8-Lip compare to sLIP?

4. After the above study, in silico prediction of peptide interaction with the nicotinic receptor are carried out in order to identify an optimal peptidic-ligand for brain targeting, that would also have good immunocompatibility when linked to liposomes, the D8. This is supposed to have increased immunocompatibility. The authors study its stability in mouse serum (Fig 7.a) and the uptake and transcytosis of liposomes with the specific ligand D8-sLip, in vitro (Fig 7 b and c). Also they present a microscopic observation of brain sections (after in vivo injection in mice). In this last experiment they compare only with the sLip. I would like to know how the DCDX-sLip compares with these results. Also is there any quantitative result to show what they have succeeded in

terms of brain targeting in vivo? Furthermore, the physicochemical characteristics of these optimal D8-sLip are absolutely missing (unless I cannot see where they are presented), together with any indication about their stability (in vitro and in vivo, as for all the liposome-types used in the study). In Fig. 8 the immunocompatibility of D8-sLip is compared with the worst case of liposomes in the whole study (the DCDX ones). This is not a correct experimental setup. In addition to the comparison with a non-immunocompatible formulation, a more compatible one (positive control) should be also used here. I would like to see the comparison with the sLip here (and with the DCDX- for in vivo brain targeting [see also below]).

5. In Fig. 9 the authors try to make their point by comparing the net-positive charge of peptide ligands with IgM levels (a), BUT in this graph the point for D8-sLip is an outline, and this is not commented by the authors. In Fig 9.b only 3 points (in each case) are compared and linear correlation is attempted. Three points are definitely not enough to confirm any linear correlation. In order to understand the final value of the current findings, the previous questions should be answered, and added to the manuscript.

Reviewer #3 (Remarks to the Author):

The paper from Guan J et al. described the correlation between immunocompatibility and composition of protein corona on brain-targeted liposomes surface.

This is a very interesting work and the results reported represent an advance in understanding how rationally design immunocompatible nanoparticles. However, I have some concerns:

- 1- How is it possible that the liposomes size decrease after incubation with serum for all preparation with except for agiopep2-lipo? This means that no protein corona is formed on liposome surface? Please, discuss it.
- 2- The interaction between liposomes and plasma protein has been done with 50% of fresh medium. What happen with an higher amount of serum?
- 3- Why the PK has been done in rats while all the other experiments have been conducted in mice? Please, discuss this point.
- 4- No data about the potential toxicity of liposomes have been conducted.
- 5- Please add some details about the liposomes preparation e.g. how unincorporated material has been removed? how many peptide molecules are present for each liposome? Discuss why liposomes prepared by extrusion through 100-nm pore size have a diameter of 140 nm.
- 6- The in vitro experiment to assess the brain transport efficiency of liposomes have been done using 10% of FBS serum, but in this condition the corona formed on liposomes surface should be different respect to that formed in mouse serum or rat serum. How is the BBB permeability in absence of serum? Please, add some data about the tightness of the BBB model.

Reviewer #1 (Remarks to the Author):

This article describes a methodological approach to address immunocompatibility issues of surface-modified liposomes and design liposomes which are more immunocompatible. By studying the interaction of CDX peptide-modified liposomes in a step-by-step manner, the authors have been able to convince this reviewer with the data and interpretation. I have no hesitation in congratulating the investigators for this excellent study that too after a long time in the liposome field. Having said this, there are a few concerns that need attention to improve this work.

The only major regret is the absence of *in vivo* ABC data which would provide confirmation about the better immunocompatibility of the final D8-liposomes.

Response: We thank this reviewer's favorable consideration of our manuscript. To address the concern about *in vivo* ABC data of the final D8-liposomes, we re-studied the PK profiles of all liposomal formulations in BALB/c mice (also intend to address the third concern raised by the reviewer #3), since studies on immunogenicity, biodistribution and protein coronas were conducted in mice. The results were shown in Figures 3a, 3b, 8c and Table 1 of the revised manuscript, confirming better immunocompatibility of the final D8-liposomes.

1. Include a mention of route of administration in Results and Discussion section. This information will help reader.

Response: We are grateful to the reviewer for this comment, and have revised the Results and Discussion section accordingly (highlighted in red).

2. Only phagocytic activity of dendritic cells and macrophages has been shown. Whether these cells are immune-activated after incubation with liposome preparations is not shown.

Response: We thank the reviewer for this comment. During immune responses,

macrophages can be activated by classical pathway towards pro-inflammatory M1 phenotype which is required for killing pathogens or activated by alternative pathway towards anti-inflammatory M2 phenotype (*J Immunol*, 164, 6166-6173). Jones and coworkers reported that PEGylated nanoparticles were mainly ingested by M2 macrophages, while M1 macrophages decreased uptake of nanoparticles (*J Clin Invest*, 123, 3061-3073). In our experiments, CDX peptide modification could enhance phagocytosis of liposomes by macrophages (Figures 2 and S1), indicating that macrophages were activated by CDX modified liposomes (We add this description in the Results and Discussion section).

For DCs activation, we studied the expression of CD83 and CD86 (markers for immune-activation of DCs. Refs: *Int J Immunopath Pharmacol*, 24, 941-948; *J Control Release*, 159, 135-142) in DCs after incubation with liposomes for 12 h, and found that sLip (without peptide modifications) itself could significantly increase the expression of both markers (as shown in the following Figure R1). The effects of CDX modified liposomes on immune-activation of DCs remains elusive. We did not add this preliminary results in the revised manuscript, and further studies are ongoing in our lab.

Figure R1. Evaluation of dendritic cell activation using CD83 and CD86 as markers. BMDCs were incubated with PBS or sLip for 12 h and the expression of markers were labeled with antibodies. Positive cells were counted using flow cytometry. n=3, data are means \pm SDs. * p<0.05.

3. The authors should discuss the possibility that the CDX-modified liposomes may be interacting with cells with specific interaction. The use of competing free peptide or somehow justifying the absence of expression of a receptor for CDX on the cells used is required. It is notable that macrophages have been shown to express nicotinic receptors.

Response: We are grateful to the reviewer for this comment. We conducted a competitive binding assay to verify whether CDX-modified liposomes specifically interacted with macrophages. As shown in Figure S1 of the revised supplement information, pre-incubation with free ^DCDX peptide (200 μM) for 2 h did not affect phagocytosis of liposomes, indicating non-specific interaction between CDX-modified liposomes and macrophages. We add the experimental method, results and discussion in the revised manuscript and the supplement information.

4. Caveat for section of Electrostatic interaction...": The authors show that number of +ve charges are a factor, and so is the length of the peptide. But it is also possible that the position of a charge is also material, especially in specific interactions. In a way this caveat is proven to exist by the success of modified short D8 peptide where the authors manipulated the placement of charged amino acids.

Response: We agree with this reviewer that the position of a charge is also a matter, especially in specific interactions. Before we designed D8, in which the positions of two charged residues have been replaced (see Table S1), the intermediate eight amino acids of ^DCDX (termed ^DCDX-S8, Table S1) also showed very low IgM binding affinity after modification on liposomal surface (Figure 5 of the revised manuscript), excluding that the placement of charged amino acids in D8 is attributed to the better immunocompatibility of D8-modified liposomes.

5. PEG is no longer the only liposome-modifying polymer for stealth property. A mention of newer polymers should be mentioned in the introduction section. For

instance, superhydrophilic polymers (Nag et al. *J Pharm Sci* 2015). A statement on how these new developments would impact the conclusions drawn on the basis of this study is important.

Response: We thank this reviewer for the suggestion. In the present study, we mainly focus on the effect of peptide on the immunocompatibility of PEGylated liposomes. The designed D8 peptide might also benefit from newer polymers that can improve the performance of liposomes. We cite two published papers (*J Pharm Sci* 104, 114-123; *Pharmaceutics* 5, 542-569) in the Introduction section of the revised manuscript.

Minor comments:

1. In methods section, authors mentioned lipid A in the composition of liposomes. Why?

Response: Lipid A is a widely used adjuvant for liposome immunization (*J Biol Chem*, 268, 26279-26285). We prepared liposomes containing Lipid A for immunogenicity evaluation. In other experiments, all liposomes were prepared without Lipid A.

2. Figure 2c: Y axis title spelling of across.

Response: We revised the Y axis title accordingly.

Reviewer #2 (Remarks to the Author):

This is a very interesting paper on a very well-known problem that limits the potential of targeted drug delivery. The protein corona has been indeed studied by many groups and for many years, however the solution to the problem and the way to effectively manipulate the type of proteins that will be adsorbed on the surface of nanoparticles immediately after their appearance in the blood in order to preserve the targeting capability of the ligands and the stealth nature of nanoparticles (and maximize

immune-compatibility) has not been yet accomplished.

In this paper the main goal is brain targeting using ligand-targeted liposomes. The authors start with methodologies to correlate the properties of ligands with the types of proteins that are adsorbed on liposomes and their effects on the liposome immunocompatibility and blood circulation. After that they continue by using the previous conclusions, to design specific peptide ligands for brain targeting and carry out final tests to identify the targeting to the brain. However, there are several points that are missing or not discussed and in several instances the experimental setup, may not be optimal.

To be more specific:

1. The authors herein carry out a series of experiments to initially prove that the in vivo stability of a peptide ligand is important in order to be able to correlate the liposome pharmacokinetics with the IgG produced following liposome in vivo administration, and IgM adsorption of the surface of the liposomes (Figures 1-4). However they do not really explain what is happening in the case of the unstable peptide-bearing liposomes. This reviewer believes that a more in depth analysis of the particular results of the LCDX-liposomes may provide interesting insights about other aspects that may be important; but have not been considered, or have been overlooked.

Response: We thank this reviewer for the comment. As emphasized in the manuscript, we believe that stability of peptide ligand plays double-edged roles (as stated in Abstract “Stable positively charged peptide ligands may play double-edged roles in targeted delivery, preserving in vivo bioactivities for binding receptors and long-term unfavorable interactions with innate immune system.”) In Figures 1-4, we not only assessed the immunocompatibility of ^DCDX- and ^LCDX-modified liposomes, but also studied the possible factors that may be attributed to the discrepancy between stable and unstable peptide ligands. Based on the results shown in Figure 4, both ^DCDX- and ^LCDX-modified liposomes are able to efficiently absorb *in vitro* natural IgM

(containing enzyme inhibitor). *In vivo*, they also demonstrated comparable absorption of natural IgM 1 h after intravenous injection; however, natural IgM absorption by ^LCDX-modified liposomes decreased at 4 h after injection. In contrast, natural IgM absorption of ^DCDX-modified liposomes increased. It is also interesting that the plasma concentration of ^LCDX-sLip rebounded at 4 h in BALB/c mice (Figure 3a), which may also be attributed to the change of IgM absorption *in vivo*. Based on our previous report (*Angew Chem Int Ed*, 54, 3023-3027), ^LCDX on liposomal surface is instable in the presence of serum (we also confirmed the stability of free peptide in the present study, Figure 7a). Thus, we concluded that stability of peptide ligand plays double-edged roles, preserving *in vivo* bioactivities for binding receptors and long-term unfavorable interactions with innate immune system. We add relative description and discussion in the Results and discussion section.

2. For the previous they use control liposomes sLIP, consisted of HSPC/Chol and DSPE-PEG2000, and liposomes having on their surface L-CDX (unstable) peptide or D-CDX (stable peptide). There are a few important points about the liposomes, I would like to point-out: A) First of all the zeta potential of HSPC/CHOL/PEG liposomes is highly negative (Table 1), although there is no negatively charged lipid in their membrane (HSPC is a zwitterionic lipid). The only possible explanation for this may be the partial hydrolysis of HSPC to Lyso-PC (which is known to happen after 1 or 2 weeks, however PEG is known to shield the negative charge generated by hydrolysis of HSPC – see *Biochimica et Biophysica Acta* 1818 (2012) 2801–2807)

Response: We are grateful to this reviewer for the insightful comments. A) For zeta potential, PEG-DSPE (both mPEG-DSPE and peptide-PEG-DSPE) used in our formulations (also widely used in commercially available PEGylated liposomal formulations) is negatively charged. The amino group in DSPE is chemically conjugated with the carboxyl group of PEG to form a neutral amide bond, while the negatively charged phosphoryl group in DSPE is left. This is believed to be the main

reason for the negative charge of liposomes. Since liposomes were characterized after fresh preparation, hydrolysis of lipids would be negligible.

B) The “stealth” character of the sLip is not obvious from the pharmacokinetic profile presented in Fig. 3.a. At the 4 h time point, although there is about half of the initial DiI level remaining in the blood, there is almost no DiI in the liver. Where is the DiI??

Response: In the previous version of our manuscript, Figure 3a and 3b showed the PK profiles of liposomes in rats and Figure 3c showed liposomes distribution in mouse liver and spleen. In the revised manuscript, we add the PK profiles of liposomes in mouse to address the third concern raised by the reviewer #3 (it is also more reasonable since other experiments were conducted in mice). In the revised Figure 3a and 3b, liposomes in mouse demonstrated some difference to that in rat. At 4 h after injection, we do not observe significant decrease of initial DiI in sLip group. That is consistent with the result that low amount of DiI distributed in the liver and spleen of sLip.

C) The peptides are attached on the liposomes via maleimide-linked on DSPE-PEG3500, while the control liposomes contain only PEG-2000, Can this difference be connected with the anti-PEG-IgM levels of the LCDX-sLIP compared to the control sLIP [although the IgG levels are close for these two liposomes types (which is logical since the L-peptide should be hydrolysed in a few minutes in the blood).?] Additionally, it is not mentioned if the attachment yield of the different ligands is measured and how. Are all the ligands attached by 100% (perhaps this is the case if ligand-PEG-lipids are synthesized initially – however it would be good to also measure the amount of ligand on the various liposome types)

Response: As shown in Figure 1, ^LCDX-sLip demonstrated comparable IgG titer to sLip, while significant enhancement in IgM titer. It was also reported that PEG length on the liposomal surface has no effect on ABC effect (*J Control Release*, 105,

305-317), thus we believe that the PEG length in the present study would not be a matter for Anti-PEG-IgM levels. All peptide-PEG-DSPE materials were chemically synthesized by attaching peptides with mal-PEG₃₄₀₀-DSPE (¹H-NMR spectra have been added in Figure S3) before liposomes preparation. After hydration, liposomes were homogenized by extrusion through membranes of different pore sizes. This process was very smooth and normally we considered no loss of peptide modified materials.

D) Through the full manuscript, all the results associated with the physicochemical properties of the liposomes (Table 2 and 3) are reported without SD values, although it is mentioned that n=3. This makes it difficult to understand the significance of specific differences between different types of liposomes. Furthermore, the authors do not comment at all about the very dramatic increase of the PDI values when the liposomes are measured after serum incubation. If the size presented is the mean size, I wonder if there is just one peak or several (due perhaps to aggregated liposomes) in the analytical report of the Malvern sizer. Also how were non-adherent proteins removed from the liposomes? Since the liposomes are small, it would be difficult to separate them from large serum proteins by ultracentrifugation.

Response: The size presented in our manuscript was the mean size and we add SD values in the revised Tables 2 and 3. We observed the increase of PDI after incubation with serum, which would be explained by: 1) the formation of protein corona is a dynamic process, which may increase the PDI of liposomes. 2) the mixture of large plasma proteins and plasma microvesicles (such as exosomes) with liposomes increased the PDI. We did not remove non-adherent proteins from the liposomes by ultracentrifugation, because it is very easy to promote liposomes aggregation. Alternatively, we directly measured the size and size distribution of liposomes in serum without further purification, and the decrease of mean size is consistent with previous reports (*ACS Nano*, 9, 8142-8156; *Nanoscale*, 8, 6948-6957).

As shown in the following Figure R2, even though the PDI increased, all liposomes except Angiopep-2-sLip demonstrated a single peak after incubation with serum. Angiopep-2-sLip displayed two peaks (100 nm and 1000 nm), while the reason remains elusive.

We add those description and discussion in the Results and discussion section of the revised manuscript.

Figure R2. Size and distribution of liposomes without or with incubation with serum.

E) the high level of IgM after 1 h on LCDX-sLip is very strange and no explanation is given; results are more logical after 4 h. Do the authors have any information about the integrity of the different liposomes they are studying after 1 or 4 h in presence of serum proteins? Are they sure they are still intact and nano-sized? Are they aggregated? Is the Dil still associated with the liposomes?

Response: Both ^DCDX-sLip and ^LCDX-sLip absorbed much more natural IgM at 1 h than sLip, indicating that CDX peptides on liposomal surface could rapidly interact with natural IgM *in vivo*. The content of IgM in the formed protein corona of ^DCDX-sLip increased at 4 h compared with that at 1 h. On the contrary, the content of natural IgM in the formed protein corona of ^LCDX-sLip at 1 h after injection was comparable to that of ^DCDX-sLip; while it significantly decreased 4 h after injection.

This may be explained by the proteolysis of ¹LCDX in blood circulation. This has also been reflected in the PK profiles (Figure 3a). The plasma concentration of ¹LCDX-sLip decreased from 30 min to 2 h, but it rebounded at 4 h.

The integrity of liposomes was evaluated (see the revised Materials and methods Section), no leakage of DiI was observed after 4 h incubation in serum. We also did not see liposome aggregation.

3. After the first part of the study, the authors investigate the effect of the net charge of the peptide ligands on the electrostatic interaction with natural IgM, and show that a specific ligand CDX-S8 that has a specific amount of net charge, results in minimal interaction with IgM, compared to other liposome types; based on this, they then prepare a new small peptide as a ligand for brain targeting. However, in this experiment (presented in Figure 5) some of the liposomes used are very different in order to make any solid conclusions (case of DOTAP), so I don't understand why there were selected initially. Also, it is confusing that the results are presented differently in Figures 5. b and d. How does CDX-S8-Lip compare to sLIP?

Response: We thank the reviewer for this comment. In this section, we intend to understand the effect of net charge of peptide ligands on the electrostatic interaction with natural IgM. DOTAP liposome was selected as a positive control since it is highly positive charged. DOTAP liposome is much more positively charged than other peptide modified liposomes; however, DOTAP liposomes induced less natural IgM absorption than ^DCDX-liposomes, indicating the effects of positive charge of peptide ligands on natural IgM absorption is different from that of lipid. We add the discussion in the revised manuscript and our future studies would focus on this concern to understand why. To address this reviewer's concern about the results in Figure 5 b and 5 d, we added sLip group and the results were revised in the manuscript. ^DCDX-S8-sLip showed comparable IgM absorption to that of sLip.

4. After the above study, *in silico* prediction of peptide interaction with the nicotinic receptor are carried out in order to identify an optimal peptidic-ligand for brain targeting, that would also have good immunocompatibility when linked to liposomes, the D8. This is supposed to have increased immunocompatibility. The authors study its stability in mouse serum (Fig 7.a) and the uptake and transcytosis of liposomes with the specific ligand D8-sLiP , *in vitro* (Fig 7 b and c). Also they present a microscopic observation of brain sections (after *in vivo* injection in mice). In this last experiment they compare only with the sLip. I would like to know how the DCDX-sLip compares with these results. Also is there are any quantitative result to show what they have succeeded in terms of brain targeting *in vivo*? Furthermore, the physicochemical characteristics of these optimal D8-sLip are absolutely missing (unless I cannot see where they are presented), together with any indication about their stability (*in vitro* and *in vivo*, as for all the liposome-types used in the study). In Fig. 8 the immunocompatibility of D8-sLip is compared with the worst case of liposomes in the whole study (the DCDX ones). This is not a correct experimental setup. In addition to the comparison with a non-immunocompatible formulation, a more compatible one (positive control) should be also used here. I would like to see the comparison with the sLip here (and with the DCDX- for *in vivo* brain targeting [see also below]).

Response: We thank the reviewer for this comment. Figure 7d and 7e was revised to address the concern about transcytosis efficiency of ^DCDX-liposome and the quantitative evaluation *in vivo*. Characterization of D8-liposomes was added in Table 3. In Figure 8a, we added the absorption of natural IgM and also quantified it in Figure 8b. Stability of D8 was shown in Figure 7a, and that of liposomes was tested in serum after 4 h incubation (see the revised Materials and methods section).

5. In Fig. 9 the authors try to make their point by comparing the net-positive charge of peptide ligands with IgM levels (a), BUT in this graph the point for D8-sLip is an

outline, and this is not commented by the authors. In Fig 9.b only 3 points (in each case) are compared and linear correlation is attempted. Three points are definitely not enough to confirm any linear correlation.

Response: We thank the reviewer for this comment. The absorption of natural IgM can be influenced by many factors. Modification of long, stable positively charged peptide ligands on the surface of stealth liposomes was inclined to absorb natural IgM. The net-positive charge of peptide ligands is only one factor, while both charge and length have been taken into consideration in the design of D8 peptide. Thus, in this graph the point for D8-sLip is an outline. We add the comments in the revised manuscript.

Figure 9b is deleted in the revised manuscript.

Reviewer #3 (Remarks to the Author):

The paper from Guan J et al. described the correlation between immunocompatibility and composition of protein corona on brain-targeted liposomes surface.

This is a very interesting work and the results reported represent an advance in understanding how rationally design immunocompatible nanoparticles. However, I have some concerns:

1-How is it possible that the liposomes size decrease after incubation with serum for all preparation with except for agiopep2-lipo? This means that no protein corona is formed on liposome surface? Please, discuss it.

Response: We thank the reviewer for this comment. The size presented in our manuscript is the mean size. We did not remove non-adherent proteins from the liposomes by ultracentrifugation, because it is very easy to promote liposomes aggregation. Alternatively, We directly measured the size and size distribution of

liposomes in serum without further purification, and the decrease of mean size and increase of PDI are consistent with previous reports (*ACS Nano*, 9, 8142-8156; *Nanoscale*, 8, 6948-6957). The mixture of large plasma proteins and plasma microvesicles (such as exosomes) may be attributed to the decrease of the mean size of particles. We add this in the Results and discussion section of the revised manuscript.

2-The interaction between liposomes and plasma protein has been done with 50% of fresh medium. What happen with a higher amount of serum?

Response: We thank the reviewer for this comment. In this work we also studied the formed protein corona of 10% liposomes mixing with 90% mouse serum (As shown in the following Figure R3). It clearly shows that the ratio between liposomes and serum does not affect the composition of protein corona (especially the IgM band).

Figure R3. Separation of protein corona formed in 90% mouse serum and 10% liposomes (volume ratio) by SDS-PAGE. IgM (at Mw 72 kDa, embraced in red circle) was characterized by nano-LC-MS/MS.

3-Why the PK has been done in rats while all the other experiments have been

conducted in mice? Please, discuss this point.

Response: We appreciate the reviewer for this comment. We studied PK profiles in rats at the very beginning since rats are very commonly used in PK study. However, we agree with the reviewer that mice are the better choice in the present study, because all other experiments were conducted in mice. In the revised manuscript, we re-conducted the PK studies and showed the results in Figures 3a, 3b, 8c and Table 1. We also compared the PK profile of D8-sLip with other liposomal formulations. The results confirmed that CDX modified liposomes also exhibited rapid clearance in mice and D8-sLip possessed much better PK profile.

4-No data about the potential toxicity of liposomes have been conducted.

Response: We thank the reviewer for this comment. In the revised manuscript, we evaluated the cytotoxicity of D8 and D8-sLip against bEnd.3 cells (widely used for construction of in vitro BBB) and AML12 (hepatic cell lines derived from mouse). The results shown in Figure S2 of the revised supplemental information confirmed that D8 and D8-sLip are nontoxic. We add the result in the Results and discussion section of the revised manuscript.

5-Please add some details about the liposomes preparation e.g. how unincorporated material has been removed? how many peptide molecules are present for each liposome? Discuss why liposomes prepared by extrusion through 100-nm pore size have a diameter of 140 nm.

Response: We thank the reviewer for this comment. More details of the liposome preparation have been added in the Materials and methods section of the revised manuscript. In the present study, we prepared peptide modified liposomes by directly incorporation of peptide-PEG3400-DSPE (1H-NMR spectra have been added in Figure S3). After hydration, liposomes were homogenized by extrusion through membranes of different pore sizes. This process was very smooth and normally we

considered no loss of peptide modified materials. For the liposomes size, different groups have reported different sizes of liposomes after extrusion through 100-nm pore size (*J Control Release*, 238, 58-70; *J Control Release*, 218, 13-21). We re-measured the size of liposomes made by the same way, and the results were quite reproducible.

6-The *in vitro* experiment to assess the brain transport efficiency of liposomes have been done using 10% of FBS serum, but in this condition the corona formed on liposomes surface should be different respect to that formed in mouse serum or rat serum. How is the BBB permeability in absence of serum? Please, add some data about the tightness of the BBB model.

Response: We thank the reviewer for this comment. We assessed the brain transport efficiency of liposomes using 10% FBS serum, which is the essential condition for maintaining the primary brain capillary endothelial cells. In the revised manuscript, we studied the brain transport efficiency of sLip, ^DCDX-sLip and D8-sLip (Figure 7d and 7e) in BALB/c mice, and the result was consistent with that of *in vitro* study. Both ^DCDX-sLip and D8-sLip demonstrated higher brain transport than sLip.

The tightness of *in vitro* BBB monolayer was monitored by measuring the transendothelial electrical resistance (TEER > 250Ω/cm²) and the information has been added in the revised Materials and methods section.

Reviewers' comments:

Reviewer #1 (Remarks to the Author):

The authors have addressed comments in a satisfactory manner.

Reviewer #2 (Remarks to the Author):

The revised manuscript submitted by the authors, together with the rebuttal letter, successfully explain/answer most of the issues raised initially. Thereby, this very interesting work deserves to be published.

Anyhow, I still cannot understand how the specific liposome compositions give such highly negative zeta-potential measurements. I personally work with liposomes for many years and never get such values with PEGylated liposomes (the most is around -5 mV), unless there is also a charged lipid in the membranes, such as PG, PA, etc. Thereby, I believe this may be due to the particular media used to dilute the samples for measurement, which is not described analytically in the methods section, and also the lipid concentration used is not mentioned. Furthermore, it may be due to the equipment settings (can the authors mention these??).

This issue, is not very important, since the authors probably follow the same procedure for all the measurements and compare the results that were generated using identical (probably) conditions. However, in such a high impact journal, all the differences with literature values, should be somehow explained.

Reviewer #3 (Remarks to the Author):

Accepted. The answers provided and the experiments made (and results) are satisfactory.

Reviewer #2:The revised manuscript submitted by the authors, together with the rebuttal letter, successfully explain/answer most of the issues raised initially. Thereby, this very interesting work deserves to be published.

Anyhow, I still cannot understand how the specific liposome compositions give such highly negative zeta-potential measurements. I personally work with liposomes for many years and never get such values with PEGylated liposomes (the most is around -5 mV), unless there is also a charged lipid in the membranes, such as PG, PA, etc. Thereby, I believe this may be due to the particular media used to dilute the samples for measurement, which is not described analytically in the methods section, and also the lipid concentration used is not mentioned. Furthermore, it may be due to the equipment settings (can the authors mention these??).

This issue, is not very important, since the authors probably follow the same procedure for all the measurements and compare the results that were generated using identical (probably) conditions. However, in such a high impact journal, all the differences with literature values, should be somehow explained.

Response: We appreciate the reviewer's comment on the measurement of liposomes zeta-potential. PEGylated liposomes have been made for three decades, while the zeta-potentials were reported in a wide range (from ~ -5 mV to ~ -40 mV) by different groups (very similar lipid compositions. There are a lot of Refs, such as: 1) *Int J Pharm*, 2008, 356, 29-36. 2) *Biophysical J*, 1992, 61, 902-910. 3) *J Control Release*, 2011, 153, 141-148.). The fact is that many factors affect the measurement of liposome zeta-potential, at least including pH value, ion, lipid concentration, and maybe also including the machine. In our lab, we prefer to measure the size and zeta-potential of liposomes in deionized water, and the results are very reproducible. As stated by the reviewer, we characterized different populations of liposomes using the same conditions, thus those results are intercomparable. We have described the detail in the method section (subsection of "Preparation and characterization of liposomes") of the revised manuscript.

REVIEWERS' COMMENTS:

Reviewer #2 (Remarks to the Author):

OK! The paper can be accepted for publication!

Explanations provided are sufficient. The authors clearly added the conditions of the zeta-potential measurements in the revised ms. Usually water should not be used to measure liposome size and zeta, since they are initially formed in isotonic media and this may result in bursting of the liposomes. In any case, since all the measurements are carried out under identical conditions, I believe the results can be of some value.